# PEGylation of Terminal Ligands as a Route to Decrease the Toxicity of Radiocontrast Re_6_-Clusters

**DOI:** 10.3390/ijms242316569

**Published:** 2023-11-21

**Authors:** Aleksei S. Pronin, Tatiana N. Pozmogova, Yuri A. Vorotnikov, Georgy D. Vavilov, Anton A. Ivanov, Vadim V. Yanshole, Alphiya R. Tsygankova, Tatiana Ya. Gusel’nikova, Yuri V. Mironov, Michael A. Shestopalov

**Affiliations:** 1Nikolaev Institute of Inorganic Chemistry SB RAS, 3 Acad. Lavrentiev Ave., Novosibirsk 630090, Russia; pronin@niic.nsc.ru (A.S.P.); tnp_post@mail.ru (T.N.P.); ivanov338@niic.nsc.ru (A.A.I.); alphiya@niic.nsc.ru (A.R.T.); tguselnikova@niic.nsc.ru (T.Y.G.); shtopy@niic.nsc.ru (M.A.S.); 2National Medical Research Center for Circulation Pathology n.a. Academician E.N. Meshalkin, 15 Rechkunovskaya St., Novosibirsk 630055, Russia; frost20@yandex.ru; 3International Tomography Center SB RAS, 3a Institutskaya St., Novosibirsk 630090, Russia; vadim.yanshole@tomo.nsc.ru; 4Department of Physics, Novosibirsk State University, 2 Pirogova Str., Novosibirsk 630090, Russia; 5Department of Natural Sciences, Novosibirsk State University, 2 Pirogova Str., Novosibirsk 630090, Russia

**Keywords:** octahedral chalcogenide rhenium cluster, phosphine, PEGylation, cytotoxicity, acute toxicity, X-ray contrast media, angiography, computed tomography

## Abstract

The development of novel radiocontrast agents, mainly used for the visualization of blood vessels, is still an emerging task due to the variety of side effects of conventional X-ray contrast media. Recently, we have shown that octahedral chalcogenide rhenium clusters with phosphine ligands—Na_2_H_14_[{Re_6_Q_8_}(P(C_2_H_4_COO)_3_)_6_] (Q = S, Se)—can be considered as promising X-ray contrast agents if their relatively high toxicity related to the high charge of the complexes can be overcome. To address this issue, we propose one of the most widely used methods for tuning the properties of proteins and peptides—PEGylation (PEG is polyethylene glycol). The reaction between the clusters and PEG-400 was carried out in acidic aqueous media and resulted in the binding of up to five carboxylate groups with PEG. The study of cytotoxicity against Hep-2 cells and acute toxicity in mice showed a twofold reduction in toxicity after PEGylation, demonstrating the success of the strategy chosen. Finally, the compound obtained has been used for the visualization of blood vessels of laboratory rats by angiography and computed tomography.

## 1. Introduction

Since the discovery of X-ray irradiation in 1895 by Wilhelm C. Röentgen [1], it became one of the most important tools in medicine diagnostics. Due to its high penetrating power, X-rays are widely used to visualize dense tissues, while special radiopaque agents are required to obtain high-quality images of soft tissues, mainly blood vessels. X-ray contrast media are based on the compounds containing heavy atoms, since in the case of solutions, the main parameter affecting X-ray attenuation efficiency is the atomic number. To date, radiocontrast agents based on 1,3,5-triiodobenzene, such as iohexol, iopamidol, ioxaglate, iodixanol, and others, are among the most widely used pharmaceuticals for intravascular administration [2,3,4,5,6]. These compounds are quite effective due to the high atomic number (Z_I_ = 53) and relatively high X-ray attenuation of iodine. However, such iodinated agents have a wide range of side effects, including cardiovascular, allergic, and pain reactions, contrast-induced nephropathy, and contraindications for use in patients with thyroid dysfunction, renal insufficiency, diabetes, and heart failure [2,3,7,8]. Gadolinium-based contrast media representing alternative diagnostic options are also not safe, mainly due to the high toxicity of gadolinium ions that can be released from chelate complexes in the presence of competing ions [9]. The scientific community also suggests the use of metal nanoparticles (Ag, Au, Pt) or metal oxides/sulfides (WO_3_, Bi_2_S_3_, CuS, etc.) [2,6,10]. Indeed, such systems show significantly better results compared to even iodinated agents, not only because of the higher atomic number, but also because of the high density due to the bulk state of the substance. Nevertheless, dispersions of solid particles tend to aggregate under harsh conditions, which can cause various pathologies. Thus, despite the great progress that has been made in this field, the development of novel radiocontrast agents is still an important actual task.

Octahedral rhenium chalcogenide cluster complexes [{Re_6_Q_8_}L_6_]^n^ (Q = S, Se, Te; L is an organic or inorganic ligand), thanks to their unique structure, occupy an intermediate state between molecular compounds and nanoparticles. The presence of a large number of heavy atoms in the cluster core provides high radiodensity and at the same time complexes do not tend to aggregate in aqueous solutions, which makes them excellent candidates for the role of X-ray contrast agents. Complexes with a {Re_6_Te_8_} core objectively exhibit the highest contrast [11,12,13], but they are difficult to manipulate synthetically and are currently poorly studied [14,15,16,17,18]. The only example of a water-soluble cluster of this type, [{Re_6_Te_8_}(CN)_6_]^4–^, showed high cell permeability, which is undesirable for a radiopaque contrast agent [11,13,19]. Complexes with phosphine ligands are an excellent alternative because they are easily synthesized in high yields under hydrothermal reaction conditions and the Re-P bond is extremely strong, which ensures high stability of the compounds [20,21,22,23,24,25,26,27]. Moreover, the chemistry of phosphines is highly developed, allowing us to manipulate the properties of the complex by choosing the appropriate ligand. Indeed, we have recently shown the promise of Na_2_H_8_[{Re_6_Se_8_}(P(CH_2_CH_2_CONH_2_)(CH_2_CH_2_COO)_2_)_6_], formed by incomplete hydrolysis of the pro-ligand P(CH_2_CH_2_CN)_3_ [11,28,29]. However, it is difficult to control the degree of hydrolysis of P(CH_2_CH_2_CN)_3_ and, consequently, the exact composition of the compound. As an alternative, complexes with fully hydrolyzed ligands Na_2_H_14_[{Re_6_Q_8_}(P(C_2_H_4_COO)_3_)_6_] (Q = S or Se) were proposed [29,30]. Indeed, these compounds showed low cell penetration as well as moderate cytotoxicity and acute toxicity. However, the results obtained are significantly inferior to the previous cluster.

To overcome this issue and increase the biocompatibility of Na_2_H_14_[{Re_6_Q_8_}(P(C_2_H_4_COO)_3_)_6_], the PEGylation strategy was chosen here. Indeed, covalent or non-covalent binding of polyethylene glycols has been widely used to increase half-lives in the bloodstream, reduce immunogenicity, and improve the solubility and stability of biological molecules such as proteins, enzymes, or pharmaceutics [31,32,33,34]. This strategy has also been used for the stabilization of metal and metal oxide nanoparticles that have been studied as X-ray contrast agents [6,10]. Since hydroxylic groups are hardly reactive in the mild conditions required for sensitive biomolecules, PEG molecules are usually modified with appropriate functional groups to form a covalent bond with the carrier [31,32,33,34]. In turn, the high chemical stability of the clusters with phosphine ligands allows us to use the classic Fischer esterification process for modification with PEG.

Thus, in this work, PEGylation of known phosphine complexes Na_2_H_14_[{Re_6_Q_8_}(P(C_2_H_4_COO)_3_)_6_] (Q = S (1) or Se (2)) with PEG-400 was performed under mild acidic conditions in aqueous media. The obtained compounds were characterized by EDS (energy dispersive X-ray spectroscopy), UV-vis (ultraviolet–visible spectroscopy), and ^1^H NMR (nuclear magnetic resonance) spectroscopy, as well as high-resolution mass spectrometry. The influence of PEGylation on the biological effects of the cluster complexes in vitro on Hep-2 cell culture (cytotoxicity and cell penetration) and in vivo on a mouse model (acute toxicity, morphological analysis, and biodistribution) was also investigated. In view of the positive results obtained, the principal possibility of using the PEGylated complexes as radiopaque agents was demonstrated by visualization of blood vessels in rats by angiography and computed tomography.

## 2. Results and Discussion

### 2.1. PEGylation of Na_2_H_14_[{Re_6_Q_8_}(P(C_2_H_4_COO)_3_)_6_] and Characterization of x-PEG (x = 1 (S) or 2 (Se))

The process of PEGylation of Na_2_H_14_[{Re_6_Q_8_}(P(C_2_H_4_COO)_3_)_6_] complexes containing 18 carboxylic groups was carried out in a boiling aqueous solution with the addition of hydrochloric acid, i.e., by modified Fischer esterification process (Figure 1) [35,36]. It is worth noting that esterification reactions are generally not carried out in aqueous medium, since an excess of water shifts the equilibrium towards the reactants, thus slowing down or even stopping the reaction. In the case of Na_2_H_14_[{Re_6_Q_8_}(P(C_2_H_4_COO)_3_)_6_], the reaction proceeds even in excess of H_2_O, probably due to the increased electrophilicity of the carboxyl groups after coordination of the phosphine to the cluster core, which facilitates nucleophilic attack by the oxygen atom of the alcohol. HCl was chosen as a catalyst because of its high volatility providing its easy removal by evaporation, and because of the formation of nontoxic NaCl upon neutralization of acid residues.

The reaction products x-PEG were isolated by precipitation with acetone, which allowed us to remove most of the unreacted PEG. It should be noted that even four redissolution/reprecipitation cycles did not lead to the formation of solid products. The oily state of x-PEG may be due to both the change in physical properties after PEGylation and the increased solubility of the products in PEG, so that even small amounts of free polyethylene glycol may be sufficient for their dissolution. To confirm the preservation of the composition of the cluster core and the primary ligand environment, the obtained compounds were studied by EDS elemental analysis and UV-vis spectroscopy (Appendix A). According to the obtained data, the ratio of heavy elements and the shape of the absorption spectra profiles remain unchanged after the reaction. Similarity of the absorption spectra of x-PEG and corresponding [{Re_6_Q_8_}(P(C_2_H_4_COO)_3_)_6_]^16–^ further allowed us to control the concentration of PEGylated products spectrophotometrically by the Beer–Lambert law using the molar absorption coefficient obtained for initial cluster solutions (ε = 76,783) (Q = S, λ = 217 nm) and 74,880 (Q = Se, λ = 227 nm). It is worth noting that PEGylation significantly increased the solubility of the reaction products in water compared to the initial complexes, which indirectly confirms the success of the esterification process. According to FTIR spectroscopy (Appendix A), the position of the vibration bands related to the carboxylic group is independent of the chalcogen in the cluster core, so that the spectra of 1/2 and 1-PEG/2-PEG pairs are almost identical. The spectra of the initial compounds (Na_2_H_14_[{Re_6_Q_8_}(P(C_2_H_4_COO)_3_)_6_]) show vibrations that can be attributed to the combination of acid and salt bands—1720 (C=O), 1550 (COO as), 1410 (combination of COO sym and C-O-H), and 1240 (C-O) cm^−1^. After PEGylation, the disappearance of the salt-related bands is observed, while the shoulder at ~1770–1780 cm^−1^ appears. This signal can be attributed to C=O vibrations in the ester complex, confirming the successful modification of the clusters.

In addition, the composition of x-PEG was investigated by ^1^H-NMR spectroscopy (Appendix A). The obtained spectra contain broad signals related to the -CH_2_-CH_2_- fragments of PEG (~3.5–3.7 ppm) and cluster complexes (~2.4–2.7 ppm), but the observed shifts are related to the high pH sensitivity of the system and are not attributed to the PEGylation process. Nevertheless, using the integrated signal intensities (Appendix A), we can estimate the cluster/PEG ratios, which were calculated to be 1/2.65 for x = 1 and 1/3.27 for x = 2 (average PEG-400 molecule contains nine -CH_2_-CH_2_- fragments which corresponds to 36 H atoms, while clusters contain 72 H atoms in CH_2_ fragments).

Since the target compounds are not isolated in solid form and do not crystallize, the most convenient method for studying their composition is high-resolution mass spectrometry (HR-MS). Thus, aqueous solutions of x-PEG were studied using HR-MS with electrospray ionization (ESI), which is considered to be one of the mildest methods for ionization of compounds. The obtained mass spectra show a rich set of signals in the 2+ (Figure 2, Appendix A) and 3+ (Appendix A) charge regions (molecular masses of initial 1 and 2 are 2919.8 and 3303.3 g mol^−1^, respectively).

Simulation of individual signals indeed confirmed the success of PEGylation of the carboxyl groups of the initial complexes with the formation of esters. The PEG-400 molecule contains ~16–18 carbon atoms and can be described by the general formula H-(O-CH_2_-CH_2_)_n_-OH (n = 8–9). In turn, the spectra show the presence of forms containing 8–10 atoms (n = 4–5), indicating at least partial fragmentation of the polymer, which is probably associated with acidic reaction media and prolonged boiling resulting in hydrolysis of the ether bonds. In general, the spectrum shows forms of single bonded H-(O-CH_2_-CH_2_)_n_-OH chains with n varying in the range of 5–20 for 1-PEG and 4–22 for 2-PEG (Figure 2, red). Rough calculations show the binding of up to 2.5 PEG molecules to one complex, but this number may be at least twice as high (~5), taking into account the presence of signals related to chains with n = 4–5, i.e., approximately half of the chain of the initial polymer. It is interesting to note that, in addition to the signals belonging to the single bonded forms, the spectrum also shows species that have undergone double esterification, i.e., double bonded forms (Figure 2, orange). In fact, this effect is highly expected due to the absence of protection of the terminal -OH group of PEG, but it can also be beneficial by providing additional screening of the cluster. In addition, hydrated forms containing 1–3 molecules of H_2_O can be observed in the spectrum. For clarity, the forms containing 2 and 3 water molecules are not presented in Figure 2A. The full spectra and an enlarged part of it are presented in ESI (Appendix A). Thus, despite the fragmentation of the polyethylene glycol molecule and the incomplete reaction, it can be concluded that the complexes are indeed easily involved in the esterification process.

### 2.2. Cell Proliferation/Viability and Cellular Uptake

Cytotoxicity of the clusters against Hep-2 cells (human epidermoid larynx carcinoma cell line) was determined using the colorimetric MTT metabolic activity assay. The results obtained are presented in Figure 3 in comparison with the data for initial compounds [29]. Since PEGylation doubles the solubility of the compounds, we were able to increase the maximum concentration studied to 11.52 mM. Half-maximal inhibition concentrations (IC_50_) were calculated to be 4.06 mM for 1-PEG and 1.93 mM for 2-PEG, which is twice as much as the values of the initial compounds (1.8 mM for 1 and 1.07 for 2) [29], thus confirming the positive effect of clusters PEGylation. The cellular uptake of x-PEG by Hep-2 cells was studied by flow cytometry. According to the data obtained, emission was observed from 2.92% of the cells for x = 1 and from 6.79% of the cells for x = 2, indicating low penetration of the cluster complexes into the cells, which is beneficial for promising radiopaque agents.

### 2.3. Acute Toxicity, Morphological Analysis, and Biodistibution

Acute toxicity was studied on 8-week-old CBA mice. Sixty-five mice were randomly divided into groups of 5 animals per dose. The injection of x-PEG solutions (100 µL) was carried out intravenously into the lateral vein of the tail. For convenient comparison with the results obtained for the initial compounds [29], the doses of 50, 150, 250, 350, 450, and 550 mg_Re_ kg^−1^ were chosen. Saline solution was administered to control groups of mice. After the injection of the solutions, the mice were kept in a vivarium with free access to food and water. The animals were monitored for 2 weeks after injection, after which the surviving mice were removed from the experiment by cervical dislocation, and organs were taken for morphological analysis. The analysis was carried out for the organs most often susceptible to pathology with intravenous administration of substances—the heart, liver, kidneys, and spleen.

The survival rates of the mice after the treatment with x-PEG in comparison with corresponding initial clusters [29] are presented in Figure 4. When the solutions were administered in high doses (550 mg_Re_ kg^−1^ for 1-PEG and 450–550 mg_Re_ kg^−1^ for 2-PEG), rapid breathing, squeaking, and other symptoms of pain were observed in mice. Death occurred instantly, or within an hour after administration, and was accompanied by convulsions, followed by a sharp stop of breathing. In turn, animals that received lower doses up to 450 mg_Re_ kg^−1^ for 1-PEG and 350 mg_Re_ kg^−1^ for 2-PEG did not show signs of pain or discomfort for 2 weeks after administration of the substances. In addition, morphological analysis of their organs did not reveal significant differences from the organs of the control group (Appendix A). Therefore, it can be concluded that the mechanism of toxicity for these substances is most likely based on an acute dose-dependent reaction of the body immediately after single intravenous administration. This may be caused by a rapid increase in pressure, vasospasm, pain shock, or other mechanisms of instant death.

The calculated median lethal doses (LD_50_) in comparison with other known octahedral cluster complexes are summarized in Table 1. One can note that similar to the cytotoxicity study, PEGylation almost doubles the in vivo toxicity of the initial compounds, thus confirming the overall success of the strategy used here. Moreover, unlike the least toxic known cluster—Na_2_H_8_[{Re_6_Se_8_}(P(CH_2_CH_2_CONH_2_)(CH_2_CH_2_COO)_2_)_6_], the composition of which is hard to control properly [29], the presented approach is highly reproducible.

To assess the distribution of compounds in the organs, x-PEGs were administered to 8-week-old CBA mice with an average weight of 21 ± 2 g. Animals were randomly divided into 6 groups (5 experimental and 1 control) of 3 mice per group. Injection of solutions (100 µL) was carried out intravenously into the lateral vein of the tail. A dose of 350 mg_Re_ kg^−1^ was chosen as it was non-toxic for both compounds. The experimental groups included mice sacrificed at 2 h, 24 h, 72 h, 1 week, and 2 weeks after injection of the solutions. Saline was administered to the control group of mice. After sacrifice, blood, lungs, heart, spleen, brain, kidneys, and liver were collected from the mice to quantify the amount of Re using inductively coupled plasma atomic emission spectroscopy (ICP-AES) (Figure 5).

It is not surprising that 2 h postinjection, the maximum concentration of the rhenium is observed in the blood and kidneys. Since the compounds are highly soluble, their excretion through the kidneys was expected. The increased concentration of Re in lungs, heart, liver, and spleen is apparently due to the rich network of capillaries and vessels providing high blood supply and not due to the accumulation of clusters in the tissues. The longest excretion of Re is observed in the kidneys and liver, indicating partial retention of the compounds in the tissues. Such behavior has been shown for high molecular weight compounds [38]. We believe that accumulation occurs in the liver by absorption by Kupffer cells [39] and in the kidney by podocytes and mesangial cells [40] of the renal corpuscle, but more detailed study is needed to confirm this assumption.

### 2.4. Angiography and X-ray Computed Tomography in Rats

To study the radiopaque properties of the x-PEG, the solutions (500 µL) at a concentration of 130 mg_Re_ mL^−1^ were administered intravenously to 10-week-old SD (Sprague Dawley) rats with an average weight of 200 ± 22 g. Thus, the total dose obtained by the animals reached 325 mg_Re_ kg^−1^, which was determined to be non-toxic for both clusters (Figure 4). The injection of solutions was carried out into the lateral vein of the tail during which an angiogram and computed tomography scan were recorded (Figure 6). The venous system, including the large caudal vena cava, is well visualized by both methods. The place where the vena cava flows into the caudodorsal part of the right atrium is not visible, since the heartbeat does not allow a clear image to be obtained. No significant differences were observed for two types of the complexes, which is due to the relatively close radiodensity of the cluster cores containing sulfur and selenium [11,19].

## 3. Materials and Methods

All solvents and reagents were commercially available and used without additional purification. Na_2_H_14_[{Re_6_Q_8_}(P(C_2_H_4_COO)_3_)_6_] (Q = S (1) or Se (2)) was prepared according to the earlier described procedure [29,30], i.e., by the hydrothermal reaction between Na_4_[{Re_6_Q_8_}(OH)_6_] and P(C_2_H_4_COOH)_3_·HCl (130 °C, 48 h).

Energy dispersive X-ray spectroscopy (EDS) was performed using a Hitachi TM3000 TableTop SEM (Hitachi High-Technologies Corporation, Tokyo, Japan) equipped with Bruker QUANTAX 70 EDS equipment. Fourier-transform infrared spectroscopy (FTIR) spectra were recorded on a Bruker Vertex 80 (Bruker Optics, Ettlingen, Germany) as KBr disks. ^1^H NMR spectra in D_2_O were recorded on a Bruker Avance III 500 MHz spectrometer (Bruker BioSpin AG, Faellanden, Switzerland). The high-resolution electrospray mass spectrometric (HR-ESI-MS) analysis was performed at the Center of Collective Use «Mass spectrometric investigations» SB RAS with a direct injection of liquid samples via an automatic syringe pump at 180 mL h^−1^ using an electrospray ionization quadrupole time-of-flight (ESI-Q-TOF) high-resolution mass spectrometer Maxis 4G (Bruker Daltonics, Bremen, Germany). Mass spectra were recorded in positive mode within 300–3000 *m*/*z* range. The calibration was performed externally using an ESI-L calibration mix (Agilent Technologies, Santa Clara, CA, USA); the typical resolution was ca. 50,000, and the accuracy was <1 ppm. Absorption spectra in water were recorded using a Cary 60 UV-vis spectrophotometer (Agilent Technologies, Santa Clara, CA, USA).

### 3.1. PEGylation of Na_2_H_14_[{Re_6_Q_8_}(P(C_2_H_4_COO)_3_)_6_] (x-PEG, x = 1 (S) or 2 (Se))

Na_2_H_14_[{Re_6_Q_8_}(P(C_2_H_4_COO)_3_)_6_] (100 mg, 34 (Q = S (1)) or 30 (Q = Se (2)) µmol) and PEG-400 (217 µL (612 µmol, Q = S) or 191 µL (540 µmol, Q = Se)) were dissolved in double distilled water (5 mL). Next, 100 µL of 36% HCl was added, and the reaction mixture was boiled for 48 h. After the reaction was stopped, the volume of the mixture was diminished to ~1 mL using a rotary evaporator. The oily products were precipitated after the addition of the excess of acetone (~10 mL). To remove unreacted PEG, the products were redissolved in water and reprecipitated with acetone twice. To neutralize the residual HCl, the complexes were again dissolved in water, the pH of the solution was adjusted to 7 with the required amount of 0.01 M NaOH, and oily x-PEGs were precipitated with excess of acetone. The reaction yields are ~95–98%. EDS: Re/S = 6/7.9 (1-PEG) and Re/Se = 6/8.2 (2-PEG). Since no solid compounds were formed and the absorption spectra of x-PEGs do not differ from the spectra of Na_2_H_14_[{Re_6_Q_8_}(P(C_2_H_4_COO)_3_)_6_], the concentration of the x-PEG solutions was controlled spectrophotometrically by the Beer–Lambert law using the molar absorption coefficient obtained for initial cluster solutions (ε = 76783 (Q = S, λ = 217 nm) and 74880 (Q = Se, λ = 227 nm)).

### 3.2. Cell Culture

Human larynx carcinoma cell line (Hep-2) was purchased from the State Research Center of Virology and Biotechnology VECTOR and cultured in Eagle’s Minimum Essential Medium (EMEM) and Dulbecco’s Modified Eagle’s Medium (DMEM) in relation to 1:1 supplemented with a 10% fetal bovine serum under a humidified atmosphere (5% CO_2_ and 95% air) at 37 °C.

### 3.3. MTT-Assay

Hep-2 cells were seeded into 96-well plates at the concentration of 5–7 × 10^3^ cells per well and then incubated for 24 h under 5% CO_2_ atmosphere at 37 °C. The cells were treated with the aqueous solutions of 1-PEG or 2-PEG (C = 115.2 mM) with a resulting concentrations range of 0.011–11.52 mM and incubated for 48 h. The 3-(4,5-dimethylthiazol-2-yl)-2,5-diphenyltetrazolium bromide (MTT) was added to each well to achieve a final concentration of 250 μg∙mL^−1^, and the plates were incubated for 4 h. The formazan formed was then dissolved in DMSO (100 μL). The optical density was measured with a plate reader Multiskan FC (Thermo Scientific, Waltham, MA, USA) at the wavelength of 570 nm. The experiment was repeated three times on separate days. The proliferation index was calculated as follows—experimental optical density (OD) value × 100/control OD value.

### 3.4. Cellular Uptake

Hep-2 cells were seeded in six-well plates at 10^5^ cells per well and were incubated for 24 h under 5% CO_2_ atmosphere at 37 °C. Solutions of x-PEG were added to the cells at final concentrations of 0.36 mM and incubated for another 24 h. After the treatment, the cells were trypsinized and resuspended in fresh phosphate-buffered saline (PBS) with 10% FBS. Cell suspensions were analyzed using CytofexS (Beckman Coulter, Brea, CA, USA). A 375 nm excitation source was used with a 695 nm emission filter. All samples were obtained from a population of 10,000 cells.

### 3.5. Animals and Housing Conditions

The in vivo studies were approved by the Ethics Committee of Federal Research Center of Fundamental and Translational Medicine (№10/1 dated 10.05.2023). All animal procedures were carried out in accordance with the protocols approved by the Bio-ethics committee of the Siberian Branch of the Russian Academy of Sciences, recommendations for the proper use and care of laboratory animals (European Communities Council Directive 86/609/CEE), and the principles of the Declaration of Helsinki. Mice and rats were housed in stainless steel cages containing sterile paddy husk as bedding in ventilated animal rooms, with free access to water and a commercial laboratory complete food.

### 3.6. Acute Toxicity Study

Sixty-five 8-week-old CBA mice with an average weight of 20 ± 3 g for females and 30 ± 4 g for males were randomized into 13 groups (12 experimental (6 groups per complex) and 1 control), each containing 5 animals per group receiving intravenous injection of the aqueous solution of 1-PEG or 2-PEG (100 µL) at doses of 50, 150, 250, 350, 450, and 550 mg_Re_ kg^−1^, or sham-injected with the same volume of saline. Animals were regularly observed for clinical signs for two weeks. The following parameters were evaluated: clinical signs and mortality (daily checks), body weight, and food and water consumption.

### 3.7. Morphological Analysis

Tissues of heart, liver, kidneys, and spleen recovered from the necropsy were fixed in 10% formalin, embedded in paraffin, sectioned, and stained with hematoxylin and eosin (HE) for histological examination using standard techniques. After HE staining, the slides were observed and photos were taken using an optical microscope (AxioImager 40, Carl Zeiss, Jena, Germany).

### 3.8. Biodistribution of x-PEG in CBA Mice

Thirty-three 8-week-old CBA mice with an average weight of 21 ± 2 g were randomized into 11 groups (10 experimental (5 groups per complex) and 1 control), each containing 3 mice. The aqueous solution of 1-PEG or 2-PEG (100 µL) was injected at a dose of 350 mg_Re_ kg^−1^ into the lateral vein of the tail. Animals were sacrificed by cervical dislocation at 2 h, 24 h, 72 h, 1 week, and 2 weeks after injection. The blood samples were collected using a standard ocular vein blood collection technique and samples of different inner organs (lungs, heart, spleen, brain, kidneys, and liver) from all the mice were obtained and weighted for the Re determination. The samples were digested at 120 °C in 1 mL of a mixture of HNO_3_ and H_2_O_2_ (3:1 *v*/*v*) for 2 h, until the solution became transparent. Deionized water was then added to 5 mL before the Re determination. The Re concentration was measured by inductively coupled plasma atomic emission spectrophotometry (ICP-AES), and the concentrations obtained were normalized to 1 g of weight of the tissue sample.

### 3.9. Angiography and X-ray Computed Tomography

Four female 10-week-old SD (Sprague Dawley) rats weighting 200 ± 22 g were anesthetized with propofol according to the manufacturer’s instructions and injected with 500 µL of aqueous solution of 1-PEG or 2-PEG with the concentration of 130 mg_Re_ mL^−1^ via the lateral vein of the tail, which corresponds to the total dose obtained 325 mg_Re_ kg^−1^. Angiogram and computed tomography scan were performed during the administration of the solution. The experiment was performed on a Precision SMART (Small animal Radiation Therapy) system. The scanning mode—8 mA, 40 kV. The 3D images were reconstructed in the Radiant Dicom Viewer program.

## 4. Conclusions

In conclusion, in this work, the PEGylation (PEG-400) of the known octahedral chalcogenide rhenium cluster complexes with phosphine ligands Na_2_H_14_[{Re_6_Q_8_}(P(C_2_H_4_COO)_3_)_6_] having 18 carboxyl groups was carried out. The reaction was conducted under simple conditions in solution in H_2_O in the presence of HCl as a catalyst and boiling. It has been shown that no change in the composition of the cluster core and the primary ligand environment occurs during the reaction. The binding of PEG molecules to the carboxyl groups of the complex with the formation of esters was confirmed by high-resolution electrospray ionization mass spectrometry. It was also shown that partial fragmentation of polyethylene glycol molecules occurs during the reaction with loss of up to half of the polyalcohol chain. Thus, up to five molecules of the polymer can be bound to each complex. The study of cytotoxicity on Hep-2 cells and acute toxicity on CBA mice showed a twofold decrease in toxicity compared to the initial complexes, while no pathologies were found in the internal organs of the animals at non-toxic concentrations. The biodistribution study showed that the compounds were mainly eliminated through kidneys and liver. Thus, in this work, we have shown that PEGylation of external ligands is a suitable way to regulate the toxicity of cluster complexes, which is one of the most important parameters for X-ray contrast agents. Here, this method allowed us to halve the toxicity of already moderately toxic initial complexes, which, in combination with no change in X-ray absorption efficiency, allows us to increase the drug concentration twice without negative effects. Indeed, the principal possibility of using PEGylated clusters as radiopaque agents was demonstrated by visualization of the laboratory rat vein by angiography and computed tomography.

## Figures and Tables

**Figure 1 ijms-24-16569-f001:**
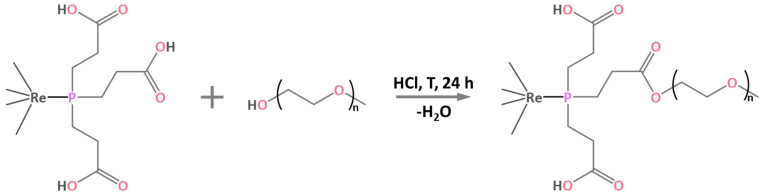
Schematic representation of the esterification of the apical ligand of the cluster.

**Figure 2 ijms-24-16569-f002:**
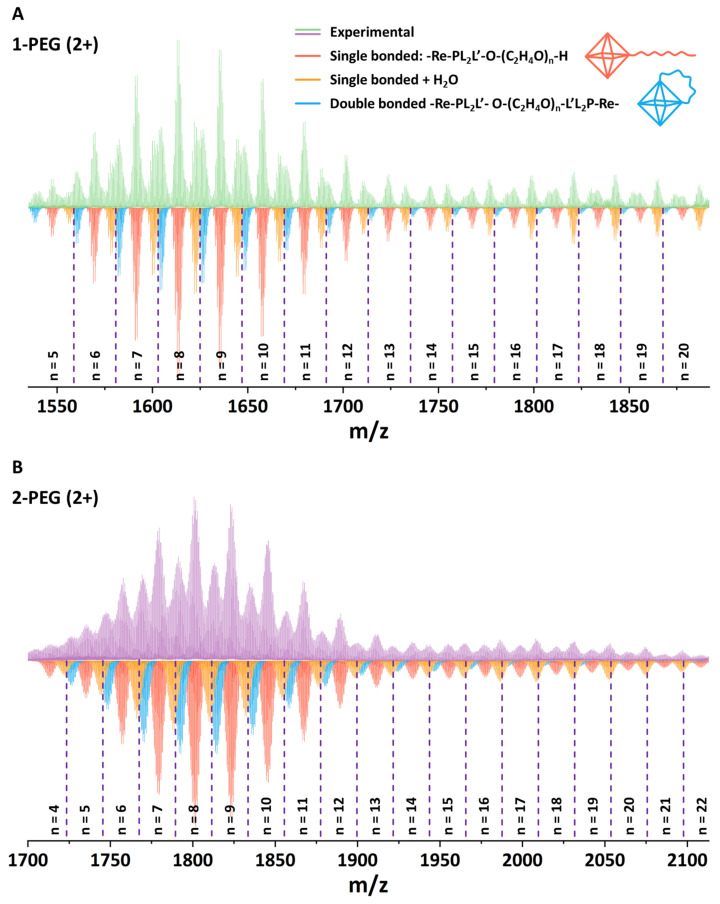
HR-ESI-MS spectra of the aqueous solutions of 1-PEG ((**A**), green) and 2-PEG ((**B**), pink) in positive mode (2+) and a simulation of PEGylated cluster forms (colored).

**Figure 3 ijms-24-16569-f003:**
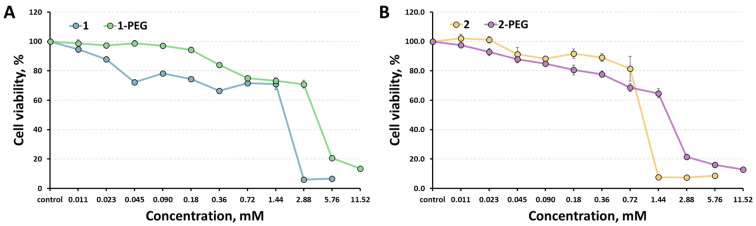
Cytotoxicity of 1-PEG (**A**) and 2-PEG (**B**) against Hep-2 cells in comparison with the corresponding initial clusters.

**Figure 4 ijms-24-16569-f004:**
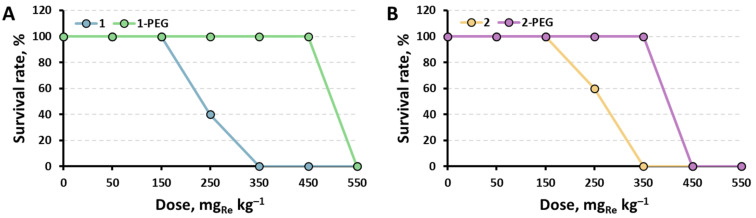
Survival rate of mice injected intravenously with 100 µL of 1-PEG (**A**) or 2-PEG (**B**) at different doses (n = 5).

**Figure 5 ijms-24-16569-f005:**
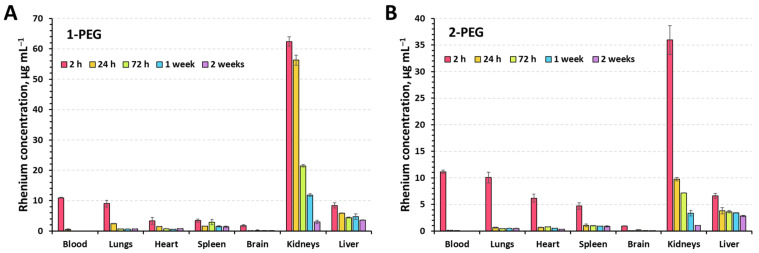
Biodistribution of the rhenium in mice organs (dose = 350 mg_Re_ kg^−1^) for 1-PEG (**A**) and 2-PEG (**B**) determined using ICP-OES.

**Figure 6 ijms-24-16569-f006:**
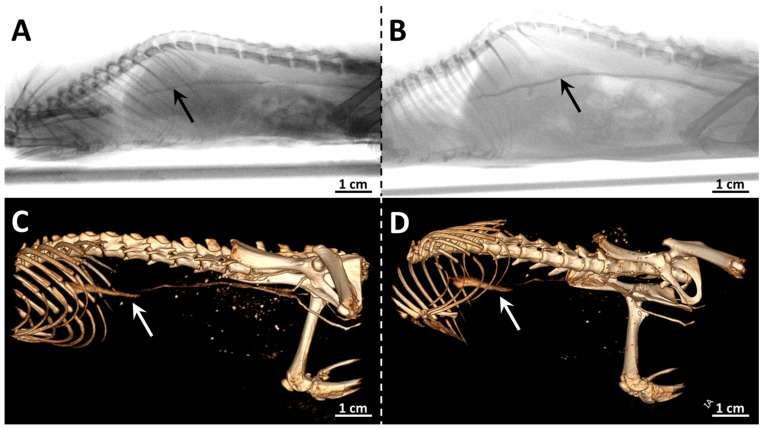
Angiography (**up**) and X-ray computed tomography images (**down**) of SD (Sprague Dawley) rats obtained during the administration of 1-PEG (**A**,**C**) or 2-PEG (**B**,**D**) (dose = 325 mg_Re_ kg^−1^). Contrasted vessels are marked with arrows.

**Table 1 ijms-24-16569-t001:** Median lethal doses (LD_50_) for x-PEG and other known octahedral rhenium clusters.

Compound	LD_50_, mg_Re_ kg^−1^	Refs.
K_4_[{Re_6_S_8_}(CN)_6_]	252	[37]
Na_4_[{Re_6_Te_8_}(CN)_6_]	504	[13]
Na_2_H_8_[{Re_6_Se_8_}(P(CH_2_CH_2_CONH_2_)(CH_2_CH_2_COO)_2_)_6_]	1589	[28]
1	256 ± 25	[29]
2	232 ± 24	[29]
1-PEG	500 ± 14	This work
2-PEG	400 ± 12	This work

## Data Availability

The data that support the findings of this study are available from the corresponding author upon reasonable request.

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
