# Peer review of "PEGylation of Terminal Ligands as a Route to Decrease the Toxicity of Radiocontrast Re6-Clusters"

_ijms, 2023, doi:10.3390/ijms242316569_

Round 1

Reviewer 1 Report

Comments and Suggestions for Authors

The manuscript presents an interesting, clear and detailed report on an attempt to create a novel contrast agent for X-ray imaging based on PEGylation idea. In my opinion, this manuscript can be published after minor revisions. Below are my suggestions for further manuscript improvement.

-       I think one short paragraph in Conclusions would be appropriate to discuss why the obtained two-fold decrease in toxicity is relevant to the field.

-       Accordingly, it is necessary to discuss briefly the Table 1 data; namely what is wrong with the agent presented in the third row of this table. Indeed, this agent has the highest LD50 value, three-fold better than the agents proposed in this work. Without such a discussion, a question arises whether the presented results are indeed relevant.

-       Line 56: I suggest to replace “emerging task” by “important actual task”.

-       Line 76: perhaps, it is necessary to replace “moderate of toxicity” by “moderation of toxicity”.

-       Line 92: the term “EDS” should be defined. This note applies to some other cases: once the term is used for the first time, it must be defined immediately, not later in the text.

-       In my opinion, Figure S1is not properly presented (or described). Absorbance is usually measured in OD units; this measurement is absolute, not relative. If the authors still wish to present the spectra in this figure in a relative manner, then they should explicitly indicate that the curves were deliberately shifted along the Y-axis. I would prefer to compare the presented absorption spectra without such a shift.

-       Lines 237-242: I could not follow how did the authors estimate the cluster/PEG ratios. I suggest the authors to verify twice this text. If they think everything is okay, then this is just my problem; but I suspect that the general reader will have this problem as well.

-       Line 268-269: “water molecules are omitted in Figure 2A” - I would suggest to rephrase the idea. The verb “omitted” makes me rather uncomfortable; it sounds like the authors wanted to hide some problem.

-       Distance scales in Figures S7 and S8 are too small and will not be seen in a printed version of the paper. Please make them much larger.

-       Figure 6 does not have distance scales at all. Please insert them in all panels.

Comments on the Quality of English Language

English is okay with me

Author Response

Referee 1

The manuscript presents an interesting, clear and detailed report on an attempt to create a novel contrast agent for X-ray imaging based on PEGylation idea. In my opinion, this manuscript can be published after minor revisions. Below are my suggestions for further manuscript improvement.

We thank the reviewer for positive evaluation of our work and relevant suggestions, which we address below.

-I think one short paragraph in Conclusions would be appropriate to discuss why the obtained two-fold decrease in toxicity is relevant to the field.

Answer: The corresponding discussion was added in Conclusion section.

-Accordingly, it is necessary to discuss briefly the Table 1 data; namely what is wrong with the agent presented in the third row of this table. Indeed, this agent has the highest LD50 value, three-fold better than the agents proposed in this work. Without such a discussion, a question arises whether the presented results are indeed relevant.

Answer: The corresponding discussion was added in text.

-Line 56: I suggest to replace “emerging task” by “important actual task”.

Answer: Corrected.

-Line 76: perhaps, it is necessary to replace “moderate of toxicity” by “moderation of toxicity”.

Answer: The sentence was corrected as follows: “Indeed, these compounds showed low cell penetration as well as moderate cytotoxicity and acute toxicity.”

-Line 92: the term “EDS” should be defined. This note applies to some other cases: once the term is used for the first time, it must be defined immediately, not later in the text.

Answer: Abbreviations were specified throughout the text.

-In my opinion, Figure S1 is not properly presented (or described). Absorbance is usually measured in OD units; this measurement is absolute, not relative. If the authors still wish to present the spectra in this figure in a relative manner, then they should explicitly indicate that the curves were deliberately shifted along the Y-axis. I would prefer to compare the presented absorption spectra without such a shift.

Answer: Due to the similar profile shape, it was difficult to compare the spectra on one graph, so we decided to shift one of the spectra up. To make the illustration clearer, each spectrum is now shown individually.

-Lines 237-242: I could not follow how did the authors estimate the cluster/PEG ratios. I suggest the authors to verify twice this text. If they think everything is okay, then this is just my problem; but I suspect that the general reader will have this problem as well.

Answer: Indeed, the ratios were calculated incorrectly. We modified the text with new calculations. Also, 1H NMR spectra with specified integral values are added in ESI for better understanding of calculations.

-Line 268-269: “water molecules are omitted in Figure 2A” - I would suggest to rephrase the idea. The verb “omitted” makes me rather uncomfortable; it sounds like the authors wanted to hide some problem.

Answer: The sentence was rephrased as follows: “For clarity, the forms containing 2 and 3 water molecules are not presented in Figure 2A.” Also, in the next sentence, we mention that the full spectra containing these forms are presented in ESI (Figures S4-S5).

-Distance scales in Figures S7 and S8 are too small and will not be seen in a printed version of the paper. Please make them much larger.

Answer: The scales were enlarged.

-Figure 6 does not have distance scales at all. Please insert them in all panels.

Answer: The scales were added.

Reviewer 2 Report

Comments and Suggestions for Authors

The paper entitled “PEGylation of terminal ligands as a route to low toxic radiocontrast Re6-clusters” contains interesting and valuable results about synthesizing pegylated derivatives of a radiocontrast Re-cluster.

The paper is logically built, and the selection of methods used during the investigations and evaluation of the spectroscopic and biological data are correct. The pegylation as a tool is useful, and gave good results, which may be/should be developed further, decreasing the toxicity further. There are, however, some remarks, which I suggest taking into consideration to improve the manuscript. 

1., The title contains “low-toxic….”. The product is not low toxic, but much less toxic than the precursor complex. The title should be changed according to this.

 2., To evaluate the MS spectra by the readwers, it had better give the molar mass of the starting cluster as well, in the text. Discussung the  distribution of the derivatives with various numbers of CH2CH2O groups characteristics for the starting PEG-400 should be given in more detail. The hydrolysis of ether bond is not as easy, so the difference between the PEG chain lengths in the products may be attributed due to other factors, e.g. differences between the reactivity of each polymer chain with different length towards the S or Se compound. 

3., The evaluation of monosubstitution/Re is probably correct, but it is not given enough detailed NMR spectral data in the ESI. The integral values are missing, which should be shown to evaluate the cluster CH2/peg CH2 group ratio. There are 72 H atoms in CH2 groups in a cluster core,  and the PEG400 mainly consist of in average 8-10 CH2CH2O groups (6x(32-40) H atom). It should be shown to confirm, that only 6X1 PEG chains are coupled in average to the cluster, and because this is an important task, I would put the NMRs into the paper.     

4., Please compare the IR spectrum of PEG400, Re-cluster, and pegylated Re cluster.  The IR spectrum of PEG 400 does not contain bands in the antisymmetric stretching C=O mode region, thus a lot of valuable information could be derived from some simple IR measurement. First of all, the wavenumber of antisymmetric  (and symmetric)  C=O bands in esterified and acidic groups probably are separated, but at least they can be separated with curve analysis.   The intensity integral ratio of esterified/free acidic carboxylates could confirm the NMR results about the pegylation degree.  

The difference between the wavenumbers of the acidic carboxylate groups symmetric and antisymmetric stretching mode bands correlates with the bonding mode of these carboxylates (free, coordinated, hydrogen bound, bidentate, monodentate, chelated, etc.), the changes/lack of changes in the coordination mode of carboxylate/carboxylic  groups in the cluster during pegylation could be followed.  It can be used only, if the PEG400 bands and symmetric C=O bands can be separated. The PEG 400 has weak IR bands in the region of symmetric C=O modes, but the authors measured NMR in D2O, thus the IR of the deuterated sample might give a chance to distinguish the PEG bands which are insensitive towards deuteration and the more deuteration sensitive carboxylic acid symmetric stretching modes. Furthermore, the deuteration can give some new pieces of information about the strength of hydrogen bonds and their changes during pegylation due to the difference between the strength of H-bonds with protium and deuterium, thus the positions/shift of O-H/O-D bands in carboxylates are sources of potential new results.  

Author Response

Referee 2

The paper entitled “PEGylation of terminal ligands as a route to low toxic radiocontrast Re6-clusters” contains interesting and valuable results about synthesizing pegylated derivatives of a radiocontrast Re-cluster.

The paper is logically built, and the selection of methods used during the investigations and evaluation of the spectroscopic and biological data are correct. The pegylation as a tool is useful, and gave good results, which may be/should be developed further, decreasing the toxicity further. There are, however, some remarks, which I suggest taking into consideration to improve the manuscript.

We thank the reviewer for positive evaluation of our work and relevant suggestions, which we address below.

  1. The title contains “low-toxic….”. The product is not low toxic, but much less toxic than the precursor complex. The title should be changed according to this.

Answer: The title was modified as follows: “PEGylation of terminal ligands as a route to decrease the toxicity of radiocontrast Re6-clusters”.

  1. To evaluate the MS spectra by the readers, it had better give the molar mass of the starting cluster as well, in the text. Discussing the distribution of the derivatives with various numbers of CH2CH2O groups characteristics for the starting PEG-400 should be given in more detail. The hydrolysis of ether bond is not as easy, so the difference between the PEG chain lengths in the products may be attributed due to other factors, e.g. differences between the reactivity of each polymer chain with different length towards the S or Se compound.

Answer: Molecular masses were added in text prior the MS spectra discussion. We believe that despite ether bonds are strong, under the selected synthetic conditions they can undergo partial cleavage, since the reaction proceeds in a strongly acidic medium in the presence of hydrochloric acid (the complex also contains 14 acid groups) at boiling for 48 hours. Also, according to the results, there are no significant differences in the reaction depth in the case of S or Se compound.

  1. The evaluation of monosubstitution/Re is probably correct, but it is not given enough detailed NMR spectral data in the ESI. The integral values are missing, which should be shown to evaluate the cluster CH2/peg CH2 group ratio. There are 72 H atoms in CH2 groups in a cluster core, and the PEG400 mainly consist of in average 8-10 CH2CH2O groups (6x(32-40) H atom). It should be shown to confirm, that only 6X1 PEG chains are coupled in average to the cluster, and because this is an important task, I would put the NMRs into the paper.

Answer: Thank you for the comment! We noticed, that the ratios were calculated incorrectly. We modified the text with new calculations. Also, 1H NMR spectra with specified integral values are added in ESI for better understanding of calculations.

  1. Please compare the IR spectrum of PEG400, Re-cluster, and pegylated Re cluster. The IR spectrum of PEG 400 does not contain bands in the antisymmetric stretching C=O mode region, thus a lot of valuable information could be derived from some simple IR measurement. First of all, the wavenumber of antisymmetric (and symmetric) C=O bands in esterified and acidic groups probably are separated, but at least they can be separated with curve analysis. The intensity integral ratio of esterified/free acidic carboxylates could confirm the NMR results about the pegylation degree. The difference between the wavenumbers of the acidic carboxylate groups symmetric and antisymmetric stretching mode bands correlates with the bonding mode of these carboxylates (free, coordinated, hydrogen bound, bidentate, monodentate, chelated, etc.), the changes/lack of changes in the coordination mode of carboxylate/carboxylic groups in the cluster during pegylation could be followed. It can be used only, if the PEG400 bands and symmetric C=O bands can be separated. The PEG 400 has weak IR bands in the region of symmetric C=O modes, but the authors measured NMR in D2O, thus the IR of the deuterated sample might give a chance to distinguish the PEG bands which are insensitive towards deuteration and the more deuteration sensitive carboxylic acid symmetric stretching modes. Furthermore, the deuteration can give some new pieces of information about the strength of hydrogen bonds and their changes during pegylation due to the difference between the strength of H-bonds with protium and deuterium, thus the positions/shift of O-H/O-D bands in carboxylates are sources of potential new results.

Answer: Thank you for your valuable comment! FTIR can indeed serve as a powerful tool to qualitatively and quantitatively determine the progress of this reaction. According to the reviewer's comment, we have recorded FTIR spectra for PEG, starting complexes, and PEGylated products. Since the ligands in the starting complexes are partially in the form of sodium salt, we see a combination of vibrations related to the acid and salt in the spectra. After PEGylation, the salt signal disappears and a low intensity shoulder near the main C=O vibration appears. Since the degree of PEGylation is small, we attributed this shoulder to the C=O oscillation in the ester. Due to the strong overlap, quantitative determination of the ratio is not possible. Corresponding discussion was added in the main text. The study of deuterated compounds by FTIR does sound interesting, but performing such studies is rather a topic for a separate independent study, which we will address in the future.
